# Direct Application of 3-Maleimido-PROXYL for Proving Hypoalbuminemia in Cases of SARS-CoV-2 Infection: The Potential Diagnostic Method of Determining Albumin Instability and Oxidized Protein Level in Severe COVID-19

**DOI:** 10.3390/ijms24065807

**Published:** 2023-03-18

**Authors:** Ekaterina Georgieva, Vasil Atanasov, Rositsa Kostandieva, Vanya Tsoneva, Mitko Mitev, Georgi Arabadzhiev, Yovcho Yovchev, Yanka Karamalakova, Galina Nikolova

**Affiliations:** 1Department of “General and Clinical Pathology, Forensic Medicine, Deontology and Dermatovenerology”, Medical Faculty, Trakia University, 11 Armeiska Str., 6000 Stara Zagora, Bulgaria; 2Department of “Medical Chemistry and Biochemistry”, Medical Faculty, Trakia University, 11 Armeiska Str., 6000 Stara Zagora, Bulgaria; 3Forensic Toxicology Laboratory, Military Medical Academy, 3 G. Sofiiski, 1606 Sofia, Bulgaria; 4Department of Propaedeutics of Internal Medicine and Clinical Laboratory, Medical Faculty, Trakia University, 11 Armeiska Str., 6000 Stara Zagora, Bulgaria; 5Department of “Diagnostic Imaging”, University Hospital “Prof. Dr. St. Kirkovich”, 6000 Stara Zagora, Bulgaria; 6Department of “Surgery and anesthesiology”, University Hospital “Prof. Dr. St. Kirkovich”, 6000 Stara Zagora, Bulgaria

**Keywords:** COVID-19, hypoalbuminemia, TEMPOL, 3-maleimido-PROXYL, ROS, EPR

## Abstract

Oxidative stress and the albumin oxidized form can lead to hypoalbuminemia, which is a predisposing factor for reduced treatment effectiveness and an increased mortality rate in severe COVID-19 patients. The aim of the study is to evaluate the application of free radical 3-Maleimido-PROXYL and SDSL-EPR spectroscopy in the in vitro determination of ox/red HSA in serum samples from patients with SARS-CoV-2 infection. Venous blood was collected from patients intubated (pO_2_ < 90%) with a positive PCR test for SARS-CoV-2 and controls. At the 120th minute after the incubation of the serum samples from both groups with the 3-Maleimido-PROXYL, the EPR measurement was started. The high levels of free radicals were determined through the nitroxide radical TEMPOL, which probably led to increased oxidation of HSA and hypoalbuminemia in severe COVID-19. The double-integrated spectra of 3-Maleimido-PROXYL radical showed a low degree of connectivity due to high levels of oxidized albumin in COVID-19 patients. The low concentrations of reduced albumin in serum samples partially inhibit spin-label rotation, with A_max_ values and ΔH_0_ spectral parameters comparable to those of 3-Maleimido-PROXYL/DMSO. Based on the obtained results, we suggest that the stable nitroxide radical 3-Maleimido-PROXYL can be successfully used as a marker to study oxidized albumin levels in COVID-19.

## 1. Introduction

The new coronavirus disease (COVID-19) is at the root of one of the most threatening pandemics in world history, causing severe acute respiratory syndrome in 10–20% of people [1]. In elderly patients or those with concomitant chronic diseases, the infection presents with severe clinical manifestations such as pneumonia, acute respiratory distress syndrome (ARDS), and multiple organ failure. Characteristic changes in the number of white blood cells, neutrophils, lymphocytes, platelets, D-dimer, and creatinine are observed [2], as is a decrease in serum albumin concentrations, which is considered to be a characteristic sign of a severe course and does not depend on the presence of concomitant diseases [3,4]. Determining the severity of the disease at an early stage is of utmost importance and will allow for timely and adequate treatment before the condition becomes irreversible. 

Pro-inflammatory molecules generated under oxidative stress (OS) cause inflammation, which plays a major role in the development of various diseases such as vascular disorders, neurodegenerative diseases, respiratory diseases, etc. [5], including acute lung damage after coronavirus infection [6]. Inflammatory reactions in the human body cause increased production of reactive oxygen species (ROS) such as superoxide anion radicals, hydrogen peroxide, hydroxyl radicals, and reactive nitrogen species (RNS) [7]. For example, in response to inflammatory stimuli, mitochondrial-generated ROS (mitoROS) are involved in the regulation of inflammatory signaling, leading to the synthesis of proinflammatory cytokines and their release from cells [8,9]. ROS and high levels of OS disrupt the structure and compromise the function of DNA, RNA, membrane phospholipids, and proteins [10].

Human serum albumin (HSA) is the most abundant transport protein in blood plasma and is responsible for the reversible binding of many endogenous and exogenous compounds. It has a high total binding capacity and is characterized as a major transporter of neutral lipophilic and acidic dosage forms [11]. The mechanism involves the generation of electrophilic products that form adducts with proteins, one of their main targets being albumin. Electrophilic profiles receive electrons from electron-rich donor molecules (nucleophiles) such as cysteine (Cys-34), thiol, and amino groups of His, Trp, and Lys, with HSA-Cys-34 being the most abundant and reactive nucleophile in human blood serum [12]. In the acute COVID-19 phase, HSA acts as an antioxidant, but high levels of free radicals can lead to irreversible oxidation of the protein. Clinical studies show that high OS levels, in combination with hypoalbuminemia, increase the risk of mortality in patients with COVID-19 [6]. 

Electron paramagnetic resonance (EPR) spectroscopy is a method that allows the study of oxidative status in biological objects and includes the study of levels of oxidative stress and free radicals in vivo [13]. In addition, site-directed spin labeling electron paramagnetic resonance spectroscopy (SDSL-EPR) is a universal technique that is used as an important tool for understanding the structure and dynamics of proteins under different conditions. Biological macromolecules do not have paramagnetism, and in order to be studied with EPR, it is necessary to introduce a paramagnetic probe whose EPR spectrum can provide information about the molecule to which it is attached [14]. To study the protein dynamics, the spin label must have a stable paramagnetic part, include a functional group capable of selectively reacting with a specific amino acid without disturbing its structure, and be sensitive to the reorientation of the protein molecule [15]. Depending on the structure of the macromolecules and the properties of the local environment, the method involves the use of specific spin-labels that meet these conditions. Such are the stable nitroxide radicals. 3-Maleimido-2,2,5,5-tetramethyl-1-pyrrolidinyoxy free radical (3-Maleimido-PROXYL, 5-MSL) [16], 2,2,5,5-tetramethyl-1-oxyl-3-methyl methanethiosulfonate (MTSL) [17], bifunctional derivatives [18], and others [14]. 3-Maleimido-PROXYL is often used to study the rotational mobility of proteins, and spectral parameters provide information on the rotation of nitroxide relative to the applied magnetic field [17]. 

In the present study, the application of 3-maleimido-PROXYL by SDSL-EPR was investigated to determine the levels of oxidized human serum albumin (oxHSA) and the conformational changes of the protein molecule in mechanically ventilated patients after SARS-CoV-2 infection. 

## 2. Results

### 2.1. Clinical Dates of Hemoglobin, C-Reactive Protein, and Lactate Dehydrogenase Levels in Patients with COVID-19

Routine clinical tests were performed on all patients with coronavirus infections. Table 1 contains data from clinical trials of hemoglobin (HGB), C-reactive protein (CRP), and lactate dehydrogenase (LDH).

Half of the patients studied had mild anemia, as confirmed by previous reports of anemia in almost 50% of patients with COVID-19 [19]. The acute immune response may be the basis of a defense mechanism characterized by the occurrence of inflammatory anemia. The imbalance of iron homeostasis is expressed in increased iron retention in the cells of the reticuloendothelial system and correspondingly high levels of ferritin as an acute phase reagent [1]. Significant deviations in LDH values from the reference range and elevated CRP values were observed, with high LDH levels associated with acute and severe lung damage in interstitial lung infections. After exploring the vaccination status of the patients included in the study, it was determined the patients had not been vaccinated against COVID-19.

### 2.2. Clinical Dates of Serum Albumin Level

To assess serum albumin levels, serum samples from healthy controls, a control group, and patients with COVID-19 were analyzed. The results of clinical trials on albumin levels in patients with COVID-19 and the control group are presented in Table 2.

Albumin is one of the major proteins in the human body that has anti-inflammatory activity, reducing cell damage by binding ROS and RNS. Acharya et al. [20,21,22] reported a direct relationship between low serum albumin levels, mortality, and complications in hospitalized patients with COVID-19. At the same time, patients with hypoalbuminemia had higher levels of CRP, leukocytes, LDH, and D-dimer and lower levels of lymphocytes and eosinophils [21]. 

In the control group, there were no significant deviations in albumin values from the reference range of 3.5 to 5.2 g/dL, while in patients with COVID-19, albumin levels were outside the reference range (Table 2). The mean values (±SD) for Group 1 and Group 2 were 4.43 g/dL in healthy controls and 2.19 g/dL ± 0.05 g/dL. The Group 2 data in the table reflect HSA analysis two days before exitus.

Figure 1A,B show albumin levels in healthy controls and patients with COVID-19 by sex and compare the values with a protein reference range of 3.5 to 5.2 g/dL (according to the methodology described in “Measuring of the serum albumin level” in the Materials and Methods section). Hypoalbuminemia (<3.5 g/dL) was observed in all patients with COVID-19.

On admission, patients in the intensive COVID-19 sector had serum albumin levels above 3.5 g/dL in both groups. As patients’ conditions worsen, protein levels decrease, reaching critically low values outside the reference range (3.5–5.2 g/dL), which is considered a poor prognostic marker for fatalities (COVID-19). The serum albumin concentrations were significantly decreased in the group of COVID-19 patients compared to healthy volunteers (*p* < 0.001) (Figure 1), and as the disease progressed and the condition of COVID-19 patients worsened, HSA gradually decreased for both sexes (*p* < 0.001) (Figure 2).

### 2.3. Chest X-ray in a Patient with COVID-19

Each patient tested for COVID-19 was subjected to an X-ray examination before hospitalization, which is part of the protocol for diagnosing patients with a coronavirus infection. Chest X-ray images showed the abnormal lung changes as multiple bilateral diffuse confluent areas of consolidation.

After radiography on admission for treatment, only an enhanced striated-mesh interstitial lung pattern was observed with the presence of discrete “frosted glass” changes on both sides and peripherally (Figure 3A). Control radiography performed on day 10 of hospitalization of the same patient showed significant progression of changes, manifested in the appearance of severe reticulonodular changes and bilateral massive areas of consolidation of the parenchyma, apically and basally located, mostly subpleural with a tendency to confluence (Figure 3B). Radiographs show a rapid progression of interstitial infiltrative-inflammatory changes and a rapid transition from first to fourth-degree changes in parenchymal radiographs.

### 2.4. CW-EPR and SDSL-EPR Spectroscopy

On ICU admission, patients were routinely screened for OS levels by X-band EPR. The method involves the determination of ROS by nitroxide radical 4-Hydroxy-TEMPO (TEMPOL), which is a suitable redox sensor for determining levels of ROS (O_2_^●−^, H_2_O_2_, etc.) and OS. 

As a result of the high levels of free radicals, the EPR signal intensity (Figure 4B) decreased dramatically at 10 min in severe COVID-19 patient. 

Oxidative stress levels in severe COVID-19 patients were measured by EPR spectroscopy and stable nitroxide radical TEMPOL. Six randomly selected patients were studied. The results were compared with those of healthy volunteers (Figure 5A).

As a result of high levels of free radicals, the EPR signal intensity and the double-integrated area of the nitroxide spectrum decreased dramatically at 10 min in patients with COVID-19, which was not observed in healthy volunteers. 

The results showed that OS levels measured at 10 min (>/=80%) were associated with poor survival in patients with COVID-19 and fully correlated with low albumin levels (Table 3).

The binding capacity of the 3-Maleimido-PROXYL spin probe in serum samples from healthy controls (Group 1) and patients with COVID-19 (Group 2) was studied and compared with EPR data from the 3-Maleimido-PROXYL spectrum/DMSO. The results of the spectroscopic study were compared with the clinical laboratory results for HSA levels. The study patients in Group 2 had a positive laboratory-confirmed PCR test and were diagnosed with bilateral pneumonia (oxygen saturation ≤ 93% at rest).

The use of spin labels and EPR spectroscopy allows the tracking of conformational changes in albumin [23]. The analysis of EPR spectra (Table 4) includes marking and measuring the hyperfine splitting parameter A_max_ between the external peaks a and c, as well as ΔH_0_ of the central resonance line (b) in the EPR spectrum of the nitroxide.

As a result of the free rotation of 3-Maleimido-PROXYL in DMSO solvent, three sharp and closely spaced spectral lines were observed that were almost identical in width and approximately the same intensity (Figure 6A). In the EPR spectra of healthy controls, changes in the A_max_ and ΔH_0_ values of the spin label were found due to a decrease in radical mobility (Figure 6B). At serum albumin levels above >3.5 g/dL (Group 1), the different values of A_max_ and ΔH_0_ and the change in the EPR spectrum of nitroxide compared to those of 3-Maleimido-PROXYL/DMSO are due to a high degree of specificity and binding of the radical to the HSA reduced form. 

In samples from COVID-19 patients, we observed visually similar EPR spectra and parameters that were close to those of 3-Maleimido-PROXYL/DMSO (no albumin available). In serum samples from patients with COVID-19, 3-Maleimido-PROXYL was characterized by low immobilization, which was probably due to the low concentration of reduced HSA and hypoalbuminemia (serum albumin levels < 3.5 g/dL).

The lower values of the double-integrated EPR spectra of the 3-Meleimido-PROXYL in healthy controls (3.5–5.2 g/dL) were probably due to the high level of reduced HSA and the formation of a covalent bond between the nitroxide and the -SH cysteine residue in the albumin (Figure 7). In COVID-19 patients, double integration of the area of the 3-Maleimido-PROXYL EPR signal showed a high level of free nitroxide. These results may be explained by hypoalbuminemia (>3.5 g/dL), oxidative damage to the protein molecule, significant conformational changes in the protein, and/or the increased level of oxidized HSA in COVID-19. Our results mirrored the ratio of oxidized to reduced albumin levels in severe COVID-19 patients, showing a significant correlation between the parameters of the EPR spectrum of the nitroxides and the severity of the disease.

In severe COVID-19 patients (*n* = 16, randomly selected), serum free thiol concentration was statistically significantly decreased (164.65 ± 6.11 µM, *p* < 0.05, *t*-test) compared to healthy volunteers (306.71 ± 5.74 µM) (Figure 8).

#### Correlation Analysis

The results presented in Table 4 represent the average of three independent measurements. The continuous parameters ΔH_0_, A_max_, and double-integrated EPR spectra of patients with COVID-19 and ΔH_0_, A_max_, and double-integrated EPR spectra of healthy volunteers were tested for normality by the Kolmogorov-Smirnov test (r = 0.5563; r = 0.5276; r = 0.5012; r = 0.672, *p* = 0.0000) for both sexes. As the assumption of normality was violated, statistical analysis was performed by an unpaired Student’s *t*-test or Mann-Whitney U test (*p* < 0.05000). The mean value of ΔH_0_ in male patients with COVID-19 compared to the control group was statistically significant (mean 1.74 ± 0.02 vs. 3.42 ± 0.03, *p* < 0.05, *t*-test). A statistically significant difference was observed in the mean A_max_ of COVID-19 patients (male) compared to controls (mean 32.93 ± 0.4 vs. 61.5 ± 0.6, *p* < 0.05, *t*-test) and in the female group compared to the control (mean 32.89 ± 0.3 vs. 61.7 ± 0.4, *p* < 0.05, *t*-test). The same was observed for the ΔH_0_ value in a group of female patients compared to healthy volunteers (mean 1.74 ± 0.01 versus 3.49 ± 0.02, *p* < 0.05, *t*-test). The correlation analysis of the EPR data showed a positive correlation (ΔH_0_ r = 0.99 and A_max_ r = 0.98) (Figure 9).

## 3. Discussion

Although COVID-19 manifests primarily as a respiratory infection, more in-depth studies suggest that the disease should be considered multi-systemic with numerous hematological findings. The various stages of infection are characterized by high levels of inflammatory markers such as CRP, interleukins (IL-2, IL-6, IL-7), LDH, ferritin, lymphocytopenia, thrombocytopenia, leukopenia, elevated procalcitonin, D-dimer, prothrombin, fibrinogen, and others [24]. For example, in critically ill patients, elevated lymphocytes [25] and neutrophils have been observed, which are associated with an increased risk of acute respiratory distress syndrome (ARDS) and death [26]. Initially, Hgb tends to decrease with age, with anemia being a common finding in older patients and usually indicative of serious diseases such as acute or chronic bleeding, inflammatory and neoplastic diseases, etc. [27]. Various studies have shown that half of the patients with COVID-19 have lower HGB levels [1], which was initially observed in the general population without COVID-19 for the same age range [28]. The average age of the studied patients (Group 2) was 56.7 ± 7, and in 50% of the group, mild anemia was observed. Therefore, lower HGB in patients with COVID-19 is not a typical feature and should not be considered a leading factor for the severity of the infection (Table 1). Terpos et al. [24] noted that about 40% of COVID-19 patients have elevated LDH levels, which is associated with a higher risk of ARDS and mortality. Table 1 shows the LDH and CRP levels and the corresponding reference ranges. Elevated levels of lactate dehydrogenase and C-reactive protein were observed in all patients with COVID-19 (Group 2). In patients with coronavirus infection, LDH and CRP can be considered markers of lung damage, as they reflect the degree of respiratory distress, and elevated values are associated with acute and severe lung damage [29]. Chest radiography is a useful method for tracking the various stages of COVID-19 with ARDS, and numerous studies have so far been published describing typical X-ray findings of pneumonia resulting from SARS-CoV-2 infection [30,31,32]. Reticulonodular changes are most commonly seen, expressed by a typical striped-mesh interstitial lung pattern [33]. The radiography investigations presented significant diffuse areas of parenchymal consolidation, typical change ground glass lung opacity, and enhanced reticular shading in all COVID-19 patients. The pleural spaces are free, and no liquid collections are established. In all patients, the hilus have a normal structure and are slightly hypervolemic. The described changes point to bilateral infiltrative-inflammatory changes with the presence of an interstitial and alveolar component corresponding to COVID-19 infection in the 3rd–4th degree of the changes (Figure 3B).

Many inflammatory and infectious diseases, traumatic injuries, and multiorgan failure are characterized by low levels of serum albumin [34], with more and more studies reporting hypoalbuminemia in patients with COVID-19 [21,35,36,37,38]. In SARS-CoV-2 infection, low protein levels are associated with increased disease severity and are a prerequisite for high mortality [35]. Therefore, albumin values below 3.5 g/dL can be considered a major factor in the determination of disease outcome. Hypoalbuminemia is characterized as a secondary disease and includes increased capillary permeability, decreased protein synthesis, a reduced half-life, and increased total serum albumin [39,40,41]. Huang et al. [4] noted a significant difference in albumin levels between surviving and non-surviving patients with severe COVID-19. In line with these data, our results confirm that in critically ill patients, hypoalbuminemia is observed, while serum albumin levels in healthy controls do not show deviations from laboratory reference values (Table 2).

HSA plays important metabolic roles by binding and transporting various molecules and modulating the immune response [42]. It has antioxidant properties, plays a major role in maintaining the redox balance in the body, and contains the largest thiol pool. In addition, the single free thiol (Cys34) in its molecule may be a functional biomarker for OS. Cys-34 is a particularly redox-sensitive site and one of the main targets of oxidative damage. About 80% of albumin is in its reduced form in healthy individuals, and the free -SH group from Cys-34 defines the protein as an antioxidant and can serve as a trap for ROS/RNS, thus participating in redox processes [43,44,45]. In various diseases, the ratio of ox/red HSA is significantly increased in favor of the oxidized form compared to healthy individuals [46]. OS, changes in pH, hydrodynamics, or temperature can potentiate the dimerization and oligomerization processes of the albumin molecule at physiological pH [47]. In patients with COVID-19, pH = 7.01 ± 0.06 (acidosis), suggesting a high percentage of oxidized albumin. HSA oxidation is associated with reducing HSA plasma levels and hypoalbuminemia, and high levels of oxidized protein forms are associated with predicting severe COVID-19 and a higher mortality risk [48,49].

HSA is an important hub where many physiological processes, pathologies, diagnosis, and therapy intersect in some diseases; in some, Cys-34 is highly modified and the amount of reduced albumin drops sharply. Chubarov et al. [50] demonstrated the formation of HSA dimers in solution, but their study did not confirm whether such dimers exist in body fluids. In vivo, it was established that, in the absence of other proteins and small molecules in the body fluids, HSA forms dimers at physiologically relevant concentrations and pH. The attachment of the spin-label 3-Meleimido-PROXYL to Cys-34 makes it impossible to measure the impact of post-translational modifications of Cys-34 on HSA dimerization. Labeling other suitable sites in HAS with other than 3-Meleimido-PROXYL nitroxides would allow the determination of the monomer/dimer equilibrium in future Double Electron Electron Resonance (DEER) EPR spectroscopy experiments.

Oxidative stress is one of the major factors in the pathology of COVID-19 [44] and is associated with extensive structural changes in HSA that suggest the proliferation of malfunctioning derivatives of this critical protein. Blood HSA is mostly monomeric, and under certain conditions, it can contain covalent dimers, trimers, etc. As thiol groups are particularly sensitive to oxidation, oxHSA is more than a biomarker for OS and may allow diagnosis of COVID-19 and determination of disease severity [51]. OS affects the binding properties of human serum albumin through purely conformational changes, and the abnormal generation of free radicals can lead to oxidative modification of Cys-34. ROS and RNS are thought to initiate disruption of albumin’s biological function by oxidizing and rendering the free thiol group biologically nonfunctional. The oxidation reaction passes through the formation of sulfenic, sulfinic, and stable sulfonic acids (Alb-Cys-34-S(=O)_2_OH) as a final products [52].

Hydrogen peroxide and neutrophil-mediated OS induce structural changes in HSA in critically ill patients with COVID-19 [53]. High levels of oxidants (H_2_O_2_, ONOO‾) or the one-electron oxidation of the thiol group to a thiyl radical (R^•^) with the formation of secondary radicals (R-SOO^•^, R-SO^•^) can lead to the formation of sulfenic acid and sulfinic acid after reactions with H_2_O_2_. As a result of albumin oxidation, the formation of dimeric disulfides is not excluded [54] during the generation of sulfenic acid as an intermediate in redox modulation. Cavalcanti et al. [55] note the role of OS in the pathogenesis of COVID-19. They unravel the role of H_2_O_2_ in critically ill COVID-19 patients and its conversion to highly reactive ROS. Because of the abnormal ROS levels (Figure 5), we assume that the % radical reduction (difference between the 3-Meleimido-PROXYL control and serum samples) is the amount of radical associated with only reduced BSA, and the double-integrated area of the EPR spectra represents a quantitative measure of the oxidized Cys-34 thiol groups (Figure 6 and Figure 8). The obtained SDSL-EPR data rather correspond to the deactivation of albumin, which is due to its irreversible oxidation to sulfinic or sulfonic acid.

On the one hand, hyperinflammation in COVID-19 may lead to reduced albumin synthesis and hypoalbuminemia, on the other hand, low albumin levels may result from high clearance of damaged and oxidized albumin [56]. Based on the obtained CW-EPR data (Figure 5), we hypothesize that in COVID-19 patients, high levels of ROS are the cause of albumin oxidation, the formation of biologically non-functional end products, and the loss of antioxidant activity. In this context, the determination of the ratio of the reduced form/oxidized form (redHSA/oxHSA) will allow an assessment of the degree of functionality and the antioxidant capacity of the protein molecule.

Although clinical and radiographic methods provide clarity regarding the current general condition of COVID-19 patients, the introduction of a new method, the SDSL-EPR, will allow additional monitoring of patients with coronavirus infection [17,32]. For this purpose, the stable nitroxide 3-Maleimido-PROXYL, which binds to albumin or other proteins, is successfully used [57]. As a result, changes in the width of the spectral lines of the spin label are observed, due to the localization of nitroxide radicals in proteins and limited rotation relative to the applied magnetic field [23]. The EPR spectra of 3-Maleimido-PROXYL in DMSO solvent showed a typical nitroxide radical triplet with characteristic small line widths [14] (Figure 6A). When studying the spectral parameters A_max_ and ΔH_0_ of the radical in serum samples from clinically healthy volunteers (Group 1), changes in the values of both parameters were observed. A high level of the reduced form of albumin probably causes limited rotation of the radical, which leads to the broadening and divergence of the spectral lines from each other, compared to the control 3-Maleimido-PROXYL/DMSO (Table 4). The anisotropy of local magnetic fields and connecting orbitals leads to a change in the position and shape of the spectra, due to changes in the dynamics of rotation of nitroxide (Figure 6B), while in solvent (DMSO), the radical has a high degree of flexibility with small line widths (Figure 6A). EPR data showed that about 90% of the albumin measured by the conventional spectrophotometric method (1.8-1.9 g/L) was in the oxidized form (Figure 6), which is probably due to oxidative damage to the protein.

Labeling of albumin with 3-Meleimido-PROXYL at its free cysteine residue allows reporting of changes in the local environment of Cys-34, with parameters a and c indicating the degree of radical immobilization and corresponding only to the Cys-34-bound nitroxide radical. The oxidized protein is less flexible and more rigid, resulting in a loss of biological activity, which makes it difficult to form a covalent bond between the spin label and the cysteine residue [23]. The measurement of the double integrated area of the EPR spectrum can be a marker in the determination of oxidized HSA in severe patients with COVID-19. High levels of OS suggest a high degree of protein oxidation, structural changes in the protein molecule, the formation of non-functional derivatives, and reduced antioxidant capacity of this critical protein in several COVID-19. Based on our results, we hypothesize that hypoalbuminemia is not only due to the inflammation associated with COVID-19 but rather to oxidative damage, which is also confirmed by other authors [51,56].

## 4. Materials and Methods

### 4.1. Chemicals

All reagents are analytically grade and were purchased from Sigma-Aldrich Corporation, St. Louis, MO, USA

### 4.2. The Patients

All patients (*n* = 66) with COVID-19 and healthy volunteers (*n* = 42) signed informed written consents from the ethics committee of the Medical Faculty and University Hospital “Prof. St. Kirkovich”. The patients were hospitalized between May 2021 and August 2022 in the Clinic of Anesthesia and Active Treatment, University Hospital “Prof. St. Kirkovich” Stara Zagora, Bulgaria. They received individual, supportive, and symptomatic treatment according to the Coronavirus Disease 2019 (COVID-19 ICU) Treatment Guidelines [https://www.covid19treatmentguidelines.nih.gov/, accessed on 13 May 2020]. Demographic (sex, age), clinical and laboratory data (Hgb, CRP, LDH, albumin), and radiographs were extracted from the available Gamma CodeMaster database 8.64.3.24809 (+3).

### 4.3. Biological Samples and Clinical Dates

The peripheral blood from patients with COVID-19 infection and controls, with a median age for each group was 56.7 ± 7 years was drawn in vacuum serum tubes. The 3-Maleimido-2,2,5,5-tetramethyl-1-pyrrolidinyloxy free radical and DMSO were purchased from Sigma-Aldrich Corporation, St. Louis, MO, USA. Additional standard 1.5 mL Eppendorf laboratory consumables and adjustable pipettes 10–100 μL and 100–1000 μL were used. Biological samples were centrifuged at 3000 rpm for 15 min at 4 °C. Biochemical screening and spectroscopic analysis were performed immediately after sampling (Figure 1).

### 4.4. Measuring of the Serum Albumin Level

For the determination of albumin concentration in human serum samples, the in vitro diagnostic reagent system Test ALB, 0-046 (COBAS INTEGRA 400/700/800, Roshe), which is based on the colorimetric method with bromcresol green (BCG), was used to determine the concentration of albumin in all-purpose samples at absorption ex/em: 629/652 nm. The method allows specific recognition of the protein without causing fluorescence in the presence of other proteins or metabolites [58,59]. It is characterized by high sensitivity and productivity and can detect serum albumin concentrations with a low detection limit of 0.08 g/dL (0–60 g/L). The assay was performed with the automatic clinical laboratory analyzer “Cobas Integra^®^ 400 Plus” (Roshe Diagnostics, Rotkreuz, Switzerland). All tested samples were measured at 18–23 °C, and the arithmetic mean of three measurements was determined.

### 4.5. Radiography of the Lungs

The tests were performed with a digital graphic X-ray machine, the PHILIPS ESSENTA DR 400, equipped with a QuantMaster Detector 3543 RG (Philips, Eindhoven, The Netherlands), characterized by high contrast resolution. A standard protocol for X-rays of the lung was applied with the following parameters: voltage 90 kV and current size 6 mAs, with light inspiration and holding the breath for 5 s. All radiographs are stored in the electronic medical records database.

### 4.6. EPR Spectroscopy

SDSL-EPR measurements of all tested samples were conducted at room temperature (18–23 °C) on an X-band EMXmicro spectrometer from Bruker BioSpin GmbH, Ettlingen, Germany, equipped with a standard resonator. Quartz capillaries were used as sample tubes. The sample tube was sealed and placed in a standard EPR quartz tube (i.d., 3 mm), which was fixed in the EPR cavity. All EPR experiments were carried out in triplicate and repeated. Spectral processing was performed using Bruker WIN-EPR and SimFonia software, Acquisit 3.0.

#### 4.6.1. EPR—TEMPOL

A solution of the nitroxide radical with a starting concentration of 2 mM was added to the serum sample in a ratio of 1:5 and stirred on a vortex for 5 s at room temperature. An aliquot of the sample is taken in a microcapillary (volume 10 µL) and placed in the EPR cavity, after which the measurement is started. For each measurement, a new amount of the reaction mixture (nitroxide/serum sample) is taken, and the analysis is performed at different incubation times (1, 10, 30, 60, 90, and 120 min). The EPR data were calculated as a percentage of the control, 0.2 mM TEMPOL/DMSO.

#### 4.6.2. SDSL-EPR Evaluation of 3-Maleimido-PROXYL 

The effect of HSA on the molecular motion of the spin label highlighted by EPR spectra was studied by the methods of Torricella and co-authors [14] and Melanson and co-authors [17], and modified by us. EPR spectra of blood serum samples from clinically healthy volunteers (*n* = 42, male *n* = 17, and female *n* = 25) were compared with those from patients with COVID-19 placed on invasive mechanical ventilation (*n* = 66, male *n* = 36, and female *n* = 30), with pO_2_ ≤ 93%, oxygen saturation, and/or PaO_2_/FiO_2_ ≤ 300 mmHg. A 200 μM solution of 3-Maleimido-PROXYL in DMSO was added to the serum samples of healthy volunteers (Group 1) and patients with COVID-19 (Group 2). After incubating the samples for 120 min at room temperature, SDSL-EPR measurements were started at 1.3 GHz, 1 mW power, 100 kHz modulation, modulation width 0.2 mT, speed 5 mT/min, and recorded at 2-min intervals.

The general EPR protocol for sample preparation and analysis involves the preparation of a 200 μM spin-stock solution in DMSO (Solution 1). Serum samples were initially incubated for about 120 min at room temperature (18–23 °C) with 3-Maleimido-PROXYL. To a 450 μL sample containing albumin, 50 μL Solution 1 was added. All samples were mixed in a vortex for 5 s (see Figure 1, step 4), and the measurement started. 

### 4.7. Measurement of Free Thiols

To prevent changes in free thiol stability, serum samples were stored at −80 °C. Serum free thiol concentrations were measured by the method of van Eijk et al. [60], with minor modifications. Serum samples were diluted 4-fold with 0.1 M Tris buffer (pH 8.2), and the background absorbance was measured at 412 nm (Thermo Scientific, Breda, Netherlands), with a reference measurement at 630 nm.

A total of 20 µL of 1.9 mM 5,5′-Dithio-bis (2-nitrobenzoic acid) (DTNB, 5,5′-Dithiobis (2-nitrobenzoic acid) from Sigma-Aldrich Corporation, St. Louis, MO, USA) was added to the samples in 0.1 M phosphate buffer (pH 7.0). We incubate for 20 min at room temperature and measure the absorbance again. Final concentrations of free thiol groups were determined by parallel measurement of a calibration curve of l-cysteine (L-Cysteine 97 52-90-4, Sigma-Aldrich Corporation, St. Louis, MO, USA) (concentration range from 15.6 µM to 1000 µM) in 0, 1 M Tris/10 mM EDTA (pH 8.2). Both intra- and inter-day coefficients of variation (CV) of free thiol measurements were <10%. Serum free thiol concentrations were adjusted for serum albumin levels by calculating the ratio of free thiol groups to albumin. This correction was made because albumin is the predominant blood protein and therefore largely determines the total amount of potentially detectable free thiol groups [61].

### 4.8. Statistical Analysis

Statistical analyses were conducted with STATISTICA 10 (StatSoft, Inc., 2011, Munich, Germany). One-way analysis of variance (ANOVA) tests were performed to compare the means of all the data. Dependent on the variance homogeneity (evaluated by Levene’s test), identification of significant differences was carried out by making use of the LSD posthoc test, while nonparametric analysis was done using the Kruskal-Wallis test. Pearson’s correlation coefficients between patient characteristics and the relative quantification of HCQ and its metabolites were calculated using Microsoft Excel (to check correlations). Comparisons between groups were performed using a Student *t*-test showing data (demographics, serum albumin date, and EPR date) as mean ± SD or Mann–Whitney U test (EPR, not normally distributed) and considered significant at *p* < 0.05. The data is presented as mean ± S.E.

## 5. Conclusions

The X-ray images of the lungs in COVID-19 patients guide the extent of damage to the lung parenchyma, and biochemical parameters showed patients in critical conditions. At the same time, the scientific community is exploring new possibilities for the diagnosis and monitoring of COVID-19 patients. According to the results presented in the current article, it may be suggested that: (1) 3-Meleimido-PROXYL is a reliable reporter of conformational changes of albumin; (2) The nitroxide radical binds molecules whose thiol group is in a reduced state, so the doubly integrated spectrum of the radical is a measure of the amount of 3-Meleimido-PROXYL bound to redHSA; and (3) The obtained biophysical parameters correlate excellently with the severity and mortality in the studied patients and justify the application of SDSL-EPR spectroscopy in COVID-19 patients. The present study shows that the EPR technique may be a promising method for monitoring severe patients with COVID-19.

SDSL-EPR has some limitations, which include: temperature (including cryogenic), pH fluctuations, reaction time, nitroxide size and dynamics, low concentration of spin-labeled radicals, etc. These disadvantages can be overcome by choosing appropriate nitroxide and methodological protocols that are adjusted to the type of medium and target protein (membranes, aqueous solution, etc.). 3-Meleimido-PROXYL is characterized as a nitroxide-specific spin-marker for labeling HSA-Cys-34 under physiological or pH-controlled conditions and manifests high stability across a wide range of conditions. Its administration leads to minimal structure-function perturbations in protein structure. Optimized SDSL-EPR analysis enables fast (scanning takes place within 2 min), reproducible, and low-cost analyses, with the only initial higher cost being the EPR spectrometer, compared to UV-Visible Spectroscopy and Colorimetric Tests. SDSL-EPR spectroscopy can solve some biologically important problems, such as the structural-dynamic information of protein systems, protein instability, and ox/redHSA, which are beyond the scope of conventional techniques.

## 6. Patents

The patents resulting from the work reported in this manuscript are identified by the incoming number of the patent application: BG/U/2022/5487.

## Data Availability

The personal data presented in this study are not publicly available due to privacy restrictions. The data are available on request from the corresponding author.

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
