# Peer review of "Direct Application of 3-Maleimido-PROXYL for Proving Hypoalbuminemia in Cases of SARS-CoV-2 Infection: The Potential Diagnostic Method of Determining Albumin Instability and Oxidized Protein Level in Severe COVID-19"

_ijms, 2023, doi:10.3390/ijms24065807_

Round 1
Reviewer 1 Report
The idea is interesting, but not at all applicable in the practice.
Firstly because of the fact that hypoalbuminemia is bad prognostic sign in a lot of diseases, it is not characteristic of COVID itself. Almost everything that you measure will be disturbed in critically ill patient.
Secondly, and even more important, why someone will use EPR for something that is so easy to measure in every laboratory and get the result in an hour.
Beside that, I have found a few misinterpretations of the literature data. For example:
First sentence in the Introduction is not correct, and citation included is not appropriate for the mortality rate of COVID. It is well known that overall mortality of COVID is not 10-20% (as stated in the manuscript), but much less. I believe that among hospitalized patients it is 10-20%, but that is not how it is stated in the first sentence of this manuscript.
etc.
Author Response
RESPONSES TO THE Reviewers' COMMENTS
We appreciate the reviewers’ comments.
Reviewer 1: The idea is interesting, but not at all applicable in the practice. Firstly because of the fact that hypoalbuminemia is bad prognostic sign in a lot of diseases, it is not characteristic of COVID itself. Almost everything that you measure will be disturbed in critically ill patient. Secondly, and even more important, why someone will use EPR for something that is so easy to measure in every laboratory and get the result in an hour.
Answer 1 (point 1): it should be noted that the present work is primary research, which should not be discounted. We fully agree that hypoalbuminemia is characteristic of many disease states and depends on age. In our article, we do not claim that the condition "hypoalbuminemia" is unique to COVID-19. Many papers indicated that low albumin levels have also been found in critically ill patients with cancer, chronic liver or kidney disease, malnutrition, sepsis and gastrointestinal bleeding, pregnancy, etc. In the article, we investigate hypoalbuminemia in patients with COVID-19, and patients with other diseases are not the subject of the study and are not included in the study. None of the patients studied had a history of liver or kidney failure, cancer, pregnancy, or bleeding. Unlike in COVID-19 patients, in those with cancer, hypoalbuminemia is due to cachexia, impaired synthesis of new albumin, increased vascular permeability, and the redistribution of albumin from the intravascular sector to the interstitium, etc. In the context of the above, several studies consider that the mechanism of hypoalbuminemia in cancer patients is malnutrition and not so much oxidative stress while redox imbalance is pronounced and develops in a short time in patients with COVID-19 (Figure 5). From the moment of entry of the SARS-CoV-2 virus, through the stage of cytokine storm and the development of the infection, the organism is subjected to acute and systemic oxidative stress, which leads to rapid and aggressive damage to important macromolecules, incl. albumin, multiple organ failure and death in a very short time. In cancer patients, moderate chronic oxidative stress is observed, and the life expectancy between cancer patients and covid patients from the moment of detection of the disease is within months or years in the first group and weeks in the second. Outside of the present study, we have conducted a detailed investigation and comparison of OS rates in patients with different stages of cancer and severe and critical COVID-19 patients. The picture of oxidative damage is unmatched, with patients with coronavirus infection in a state of "oxidative shock" that is not as dramatic as in cancer patients. For example, in gastric cancer patients, the leading cause of hypoalbuminemia is bleeding, not the high levels of oxidative stress and protein molecule damage that is a hallmark of COVID-19. Every single patient entered treatment with albumin levels in the reference range and had not previously been diagnosed with hypoalbuminemia. These data are well represented in Figure 2 (A and B), which presents a decrease in albumin levels in COVID-19 patients with a length of stay and with increasing severity of infection. Albumin levels were examined daily in all patients by a conventional spectrophotometric method as part of the standard biochemical screening, which allows monitoring of serum albumin, and the results obtained are direct evidence of the development of hypoalbuminemia as a result of COVID-19.
Answer 1 (point 2): In the scientific literature, conventional laboratory tests only examine total serum albumin levels. In order to detect oxidative-induced damages in the protein and conformational disorders in its molecule, additional analyzes should be performed (including the highly specific SDSL-EPR method). Therefore, we do not agree with your statement that the idea we present is not applicable in practice, since we should not limit ourselves to simply measuring albumin and proving hypoalbuminemia. For example, in a patient with COVID-19 and albumin levels of 1.9 g/L, 89% of the protein was found to be in the oxidized form, possibly accounting for high levels of non-functional albumin (Figure 6). This creates a prerequisite for impaired antioxidant activity of the protein and compromised drug transport. Intravenous administration of HSA will increase total protein but not improve drug transport; reaching levels of 3.5 g/L will be apparent but ineffective due to the massive oxidation of pre-infusion albumin.
Reviewer 1 (point 3): first sentence in the Introduction is not correct, and citation included is not appropriate for the mortality rate of COVID. It is well known that overall mortality of COVID is not 10-20% (as stated in the manuscript), but much less. I believe that among hospitalized patients it is 10-20%, but that is not how it is stated in the first sentence of this manuscript.
Answer 3: Done
Thank you very much that you help us to improve our manuscript. All changes in the text are colored in red.
Sincerely yours
Assoc. prof. Galina Nikolova, PhD and Head Assist. Prof. Ekaterina Georgieva, PhD
Faculty of Medicine, Trakia University, Stara Zagora, 6000 Bulgaria
Reviewer 2 Report
The authors presented the paper "Direct application of 3-Maleimido-PROXYL of proving hypoalbuminemia in cases of SARS-CoV-2 infection: The potential diagnostic method of determining albumin instability and protein level in COVID-19 critical patients."
1) The reference list should be improved. The presented above paper about the association of albumin and COVID-19, and some recent albumin review papers may be suitable for paper.
https://www.mdpi.com/1422-0067/23/18/10557
https://www.mdpi.com/2312-7481/8/2/13
https://www.mdpi.com/1422-0067/22/19/10318
https://www.mdpi.com/1422-0067/22/18/10086
The association of low serum albumin level with severe COVID-19: A systematic review and meta-analysis. Crit. Care 2020, 24, 1–4.
Review: Roles of human serum albumin in prediction, diagnoses and treatment of COVID-19. Int. J. Biol. Macromol. 2021, 193, 948–955.
Is Albumin Predictor of Mortality in COVID-19? Antioxidants Redox Signal. 2021, 35, 139–142.
Maybe, some references about Hypoalbuminemia in COVID-19 patients should be cited. At the moment, it is not clearly so new data.
2) I have found a paper
https://www.mdpi.com/2076-3921/11/12/2311
Site-Directed Spin Labeling EPR Spectroscopy for Determination of Albumin Structural Damage and Hypoalbuminemia in Critical COVID-19
Please, clearly presented the novelty of the presented work in Introduction, Results and discussion, and Conclusion sections. The part of the paper is similar.
3) Moreover, In the present work, you highlight that method show albumin concentration. However, the reaction occurs with the albumin one SH group, which is about 30% oxidized in healthy. In the pathologies, it can be 70-80% oxidized. Some covalent and non-covalent dimers and oligomers may occur https://www.mdpi.com/1420-3049/26/1/108.
In this way, this reagent reacts excellent with healthy serum, results in a spectrum broadening. I have read the papers that the albumin SH group is highly oxidized due to the oxidative stress. It is in a good correlation under COVID-19 papers. Please, present good linear correlation between albumin concentration by double integral and albumin concentration by other method in not less than 10 patients. I think that the problem may be not only in albumin level but an albumin quality. I agree that the method may works but I don't agree with the data interpretation. Moreover, it may show highly oxidized, "bad" albumin percent, which may be more important to indicate severe COVID-19
Author Response
RESPONSES TO THE Reviewers' COMMENTS
We appreciate the reviewers’ comments.
Reviewer 2
The authors presented the paper "Direct application of 3-Maleimido-PROXYL of proving hypoalbuminemia in cases of SARS-CoV-2 infection: The potential diagnostic method of determining albumin instability and protein level in COVID-19 critical patients."
Answer 1: In the present article, we hypothesize that in critically ill patients, due to high levels of oxidative stress, oxidation of the protein molecule occurs. The results showed a reduced antioxidant capacity of albumin in samples from COVID-19 patients (about 70%) compared to that of healthy volunteers, suggesting a high degree of oxidation of the protein. Taken together, these results suggest structural changes in HAS, reduced antioxidant capacity, the predominance of oxidized form, and non-functional derivatives of the protein in COVID-19 critical patients. Based on this, we hypothesize that hypoalbuminemia is not only due to the inflammation associated with COVID-19 but rather to oxidative damage, which is also confirmed by other authors' collectives.
Reviewer (point 1):
1) The reference list should be improved. The presented above paper about the association of albumin and COVID-19, and some recent albumin review papers may be suitable for paper. https://www.mdpi.com/1422-0067/23/18/10557; https://www.mdpi.com/2312-7481/8/2/13; https://www.mdpi.com/1422-0067/22/19/10318; https://www.mdpi.com/1422-0067/22/18/10086;
The association of low serum albumin level with severe COVID-19: A systematic review and meta-analysis. Crit. Care 2020, 24, 1–4.; Review: Roles of human serum albumin in prediction, diagnoses and treatment of COVID-19. Int. J. Biol. Macromol. 2021, 193, 948–955.; Is Albumin Predictor of Mortality in COVID-19? Antioxidants Redox Signal. 2021, 35, 139–142. Maybe, some references about Hypoalbuminemia in COVID-19 patients should be cited. At the moment, it is not clearly so new data.
Answer 1: In connection with new data and improvement of the article, the following authors were added:
Reviewer (point 2):
2) I have found a paper https://www.mdpi.com/2076-3921/11/12/2311. Site-Directed Spin Labeling EPR Spectroscopy for Determination of Albumin Structural Damage and Hypoalbuminemia in Critical COVID-19. Please, clearly presented the novelty of the presented work in Introduction, Results and discussion, and Conclusion sections. The part of the paper is similar.
Answer 2: The article entitled "Site-Directed Spin Labeling EPR Spectroscopy for Determination of Albumin Structural Damage and Hypoalbuminemia in Critical COVID-19" aims to present the strengths and weaknesses of conventional spectrophotometric tests and SDSL-EPR spectroscopy. In it, we compare the two methods and hypothesize that low albumin levels and structural changes in its molecule and hypoalbuminemia may impair drug transport and therapy effectiveness in critically ill patients with COVID-19. The data presented include results from a small number of patients (n=20) and represent a preliminary primary study.
Reviewer (point 3):
Moreover, In the present work, you highlight that method show albumin concentration. However, the reaction occurs with the albumin one SH group, which is about 30% oxidized in healthy. In the pathologies, it can be 70-80% oxidized. Some covalent and non-covalent dimers and oligomers may occur https://www.mdpi.com/1420-3049/26/1/108.
Natural HSA is predominantly monomeric, although under certain conditions it can form dimers, trimers, and oligomers. The formation of HSA dimers occurs at physiological pH values and at high HSA concentrations, although the self-oligomerization process can occur at or below physiological HSA concentrations. HAS dimerization and oligomerization are determined by non-covalent reversible interactions or covalent bond formation between thiol groups. Extreme conditions such as high temperatures, extreme pH values, and other high stresses potentiate the formation of non-covalent dimers and oligomers. Forming a direct disulfide bond and a covalent dimer requires a molecular distortion of the monomers to bring the Cys34 residues sufficiently close. Under physiological conditions, the dimers exit the blood vessels into the extravascular fluids, which contain much lower monomeric HSA. In healthy humans ~70% of total albumin is in the reduced form (mercaptalbumin), 25% the non-mercaptalbumin fraction (mixed disulfides between Cys34 and low molecular weight thiols), and a small fraction (~1%) oxidized thiol (sulfinic and sulfonic acids and etc.), which cannot be reduced and are end products of oxidation.
Answer 3: done
Reviewer (point 4):
In this way, this reagent reacts excellent with healthy serum, results in a spectrum broadening. I have read the papers that the albumin SH group is highly oxidized due to the oxidative stress. It is in a good correlation under COVID-19 papers. Please, present good linear correlation between albumin concentration by double integral and albumin concentration by other method in not less than 10 patients. I think that the problem may be not only in albumin level but an albumin quality. I agree that the method may works but I don't agree with the data interpretation. Moreover, it may show highly oxidized, "bad" albumin percent, which may be more important to indicate severe COVID-19
We fully agree with your comment that, in fact, changes in albumin structure may be due precisely to protein oxidation occurring as a result of high OS levels. For this purpose, we used the nitroxide radical TEMPOL. We present additional results that reflect levels of oxidative stress (OS) in 6 patients with severe COVID-19 and 6 randomly selected healthy volunteers (Figure 5).
Based on the spectra obtained from SDSL-EPR and 3-Meleimido-PROXYL, the change in shape and intensity of a and c in healthy volunteers indicates albumin with no conformational changes. In contrast, in COVID-19 patients, as a result of high and systemic oxidative stress, a and c were approximately the same as the 3-Meleimido-PROXYL/DMSO control. This is probably due to the irreversible oxidation of albumin, which creates conditions for impaired transport of drugs in the body. Simultaneously, we have used a spin label with a small bulk structure, since larger spin labels still have an increased potential to disrupt the structure of the labeled protein. The introduction of 3-Meleimido-PROXYL meets these conditions, allowing easy rotation due to small volume and obtaining a stable side chain with minimal structural-functional perturbations in the albumin molecule.
Answer 4: done
Discussion: done
Conclusion: done
References: done
Thank you very much that you help us to improve our manuscript. All changes in the text are colored in red.
Sincerely yours Dr. Galina Nikolova and Dr. Ekaterina Georgieva
Round 2
Reviewer 2 Report
Thank you for the revised paper. I have only some minor comments, which may improve paper quality.
1) Please, see the journal template. Tables look not the same. The same for Figure caption. It seems some problem with styles. I don't know why but Table 3 presented in color form.
2) Please, enlarge Figures 1, 2, 5, and 7. It is difficult to see the text. Moreover, I recommend enlarging text in all Figures. It is hard to see the scales, numbers, and text. Figures can be enlarged to the whole paper width.
3) A paper about reversible albumin dimers may be suitable for discussion section https://www.mdpi.com/1420-3049/26/1/108
4) For conclusion section, please, present some limitations or some data how much time required from blood sampling to your result data. It looks like the method is more perspective for research, neither clinics. Moreover, some comparison in price, equipment, etc. with other primary used methods should be done to present method prospects for clinics.
Author Response
Dear reviewer, Thank you very much for helping us to do our paper better.
All corrections are in red!
1) Please, see the journal template. Tables look not the same. The same for Figure caption. It seems some problem with styles. I don't know why but Table 3 presented in color form.
Answer: done
2) Please, enlarge Figures 1, 2, 5, and 7. It is difficult to see the text. Moreover, I recommend enlarging text in all Figures. It is hard to see the scales, numbers, and text. Figures can be enlarged to the whole paper width.
Answer: done
3) A paper about reversible albumin dimers may be suitable for discussion section https://www.mdpi.com/1420-3049/26/1/108
Answer: done
4) For conclusion section, please, present some limitations or some data how much time required from blood sampling to your result data. It looks like the method is more perspective for research, neither clinics. Moreover, some comparison in price, equipment, etc. with other primary used methods should be done to present method prospects for clinics.
Answer: done